# OSRT: An Online Sparse Approximation Model for Scattered Data

## Abstract

Online learning is a crucial technique for dealing with large and evolving datasets in various domains, such as real-time data analysis, online advertising, or financial modeling. In this paper, we propose a novel predictive statistical model called the Online Sparse Residual Tree (OSRT) for handling streaming multivariate scattered data. OSRT is based on online tree decomposition and online adaptive radial basis function (RBF) exploration. OSRT dynamically expands its network depth as more data arrives, and incorporates a sparse and appropriate RBF refinement at each child node to minimize the residual error from its parent node. OSRT also uses an incremental method to explore the central node of the RBF function, ensuring both sparsity and accuracy of the model. When the network reaches its maximum depth, the OSRT model updates the RBF approximation of its final layer based on the most recent data. This ensures that the model captures the latest trends in the evolving data. We evaluate our algorithm on several datasets, and compare it with existing online RBF methods. From the results, it is shown that OSRT achieves higher efficiency and accuracy.

## 1 Introduction

The research field of real-time online regression analysis addresses challenges like forecasting traffic congestion, energy usage, and even environment parameters. These tasks demand predictive modeling that operates in an online, real-time fashion and can quickly adapt to changing conditions. A common approach of solving such complex tasks is to transform them into simpler tasks, and then use known methods to solve the simple tasks. For example, the partition of unity scheme based on decomposing the original reconstruction domain into many subdomains or patches is a good choice. And it can be coupled with radial basis function (RBF) or other basic regression approach(De Marchi et al., 2019).

RBF method has shown remarkable success in multivariate scattered data approximation because of its dimensional independence and remarkable convergence properties (Buhmann, 2000; Luo et al., 2014). RBF method was originally introduced byHardy (1971) and it is an effective tool in engineering and sciences. When data is generated and collected in the form of streaming, many traditional machine learning algorithms have deriving online learning versions, RBF is no exception. And the online RBF networks have drawn significant attention in time series prediction(Liu et al., 2020b), environment parameters detectionMeng et al. (2021) and so on(Carolis et al., 2015; Scalabrini Sampaio et al., 2019).

One of the fundamental challenges of online RBF is determining the number of hidden neurons and the centers of the hidden neurons. According to the Vapnik-Chervonenkis theory[8](Cherkassky & Mulier, 1999), it is able to get a satisfied training error with the increase of network size. However, too large a network may bring in poor testing performance, leading to over-generalization. Resource-allocating network (RAN)(Kadirkamanathan & Niranjan, 1993) takes the growth strategy first, it adds RBF neurons based on the novelty of the input information. Then, pruning strategy was added into RAN, followed by the minimal resource-allocating network (MRAN)(Yingwei et al., 1997). The growing and pruning RBF(GAP-RBF) first introduces the concept of significance for the hidden neurons, and the neurons were updated based on the accuracy and significance(Huang et al., 2004; 2005). InChen et al. (2015), Chen et al. proposed a radial basis function (RBF) neural network with

a fixed number of hidden nodes, where the insignificant neurons would be replaced by new RBF neurons.

But the above methods often take structure determining and parameters optimizing as two separate tasks to tackle. Recently, researchers have been trying to address the challenge of finding a trade-off between structure size and generalization performance in the context of structure determining and parameters optimizing. One common approach is to use evolution algorithms that combine both tasks within a unified framework (Qasem et al., 2012; Alexandridis et al., 2012; Han et al., 2016). For instance, Alexandridis et al. (2012) integrated particle swarm optimization (PSO) with the fuzzy means algorithm to design a unified framework for RBF networks. Similarly, Han et al. (2016) proposed an adaptive particle swarm optimization (APSO) algorithm to simultaneously optimize network size and parameters. Yu et al. (2014) developed the incremental error correction algorithm for RBF network construction. This method gradually adds RBF nodes and adjusts parameters until the desired performance is achieved. Some researchers have also proposed hybrid constructive algorithms that simultaneously determine network size and train parameters (Wu et al., 2014; Qian et al., 2016).

The limited fitting ability of RBF networks is another challenge when the target function is too complex. It motivates us to find an approach which can simplify the target function. Tree-based models are well-suited for such tasks(Zhou & Feng, 2019). They possess the unique ability to directly split the original domain without compromising the inherent semantic meanings. Moreover, to enhance the modeling capabilities of trees, the incorporation of neural networks as predictive functions in the leaf nodes has proven beneficial (Zhou & Chen, 2002). This approach leads to the creation of a hierarchical tree-structured neural network, which exhibits expressive power at least on par with general neural networks. Consequently, the augmentation of the RBF family with flexible and potent tree-structured models appears to be a promising strategy. There have been some attempts to combine tree structures and RBF methods in the field of machine learning (Liu et al., 2020a; Akbilgic et al., 2014; Xu & Luo, 2022).

In this work, online sparse residual tree (OSRT) is developed for the purpose of representing the intrinsic structure of streaming multivariate scattered data. OSRT is based on both online tree decomposition and online adaptive radial basis function explorations. It separates the input space into smaller pieces as child nodes, in which a concise and proper RBF refinement is added to the approximation by minimizing the 2-norm of the residual inherited from its parent. And the child nodes will be further split into two according to their residual, this process stops until reaching the max depth or desired accuracy. This method draws on the algorithms of online decision tree and the methods of online RBF.

## 2 MODEL DEFINITION

In this section, we introduce the structure of online sparse residual tree. And we declare the space dimension $d \in N$, the domain $\Omega \subset R^d$ is convex, $f : \Omega \to R$ is a given target function, $X = \{x_i\}_{i=1}^N$ and $f_X = \{f(x_1), f(x_2), \ldots, f(x_N)\}$ are the input datasets.

RBF networks are a type of general linear model where the input data are transferred to feature space by non-linear transformations using RBF. There was only one hidden layer composed of RBF nodes, whose activation function is Gaussian kernel $\varphi_j(x) = e^{-\delta^2\|x-c_j\|_2^2}$, where $\delta > 0$ is often called the shape parameter and $c_j$ is the center vector of the $j^{th}$ RBF node. We consider fitting such a function $f : \Omega \to R$, where $\Omega \subset R^d$ is convex. Then the Gaussian approximation of RBF networks can be presented blow:

$$s(x) = \sum_{j=1}^N \alpha_j \varphi_j(x)$$

where $N$ is the number of hidden nodes, $\alpha_j$ is the weight connecting hidden node $j$ and output $s$.

As mentioned in Sect. 1, tree-structured functions are suited for describing numerical data. Here we use binary tree which typically divides the input space into two disjoint subsets and makes predictions using functions defined on each subset. We suppose $X \subset \Omega$ itself be the root in the beginning, and denote the domain and its relevant datasets as $\Omega_0$ and $X_0$, respectively. Suppose $\Omega_{l_j}$ is a certain convex subset of $\Omega_0$, and it represents the $j^{th}$ unit of the $l^{th}$ layer of the sparse

residual tree. And $X_{l_j} \subset \Omega_{l_j}$ represents the corresponding input data. Let the residual function $r_0(x) = f(x)$ on $\Omega_0$, then for each particular node, we need to explore the RBF approximation

$$s_{l_j}(x, \alpha) = \sum_{i=1}^{N_\chi} \alpha_i \varphi_{\delta_l}(x) \tag{1}$$

to minimize the 2-norm of the current residual

$$r_{l+1,j}(x) = r_l(x) - s_{l_j}(x), \quad \forall x \in X_{l_j} \tag{2}$$

where $N_\chi$ is the number of neurons in this node, $\delta_l$ is the shape parameter and $\varphi$ is the kernel function. Notice that $s_l$, composed of $s_{l_j}$, is only used to refine the relative global component of $r_l$ on $\Omega_l$. Finally, $r_{l+1,j}$ is combined into $r_{l+1}$ for the next layer. Most regression trees grow by recursive partitioning: the best split rule is determined by examining all cutpoint candidates along each variable. The data set is then divided according to this split rule and the process is repeated on the disjoint subsets until a stopping condition is met(He & Hahn, 2023). And a simple known regression method is conducted after the input space decomposition. When we finish building a OSRT of the target function $f(x)$ on $\Omega$, suppose the OSRT has $n_L$ leaf nodes and its $i^{th}$ leaf node domain is $\Omega_{Li}$, and we have $\bigcup_{i=1}^{n_L} \Omega_{Li} = \Omega$ and $\Omega_{Li} \cap \Omega_{Lj} = \emptyset, i \neq j$ . Then for each leaf node $X_{Li} \subset \Omega_{Li}, i = 1, 2, \ldots, n_L$, there exists one root-to-lead path. And the OSRT prediction on $\Omega_{Li}$ can be expressed as

$$s_{OSRT_{Li}}(x) = \sum_{l=0}^{L} s_l(x), \quad \forall x \in \Omega_{Li} \tag{3}$$

and the OSRT prediction on $\Omega$ is

$$s_{OSRT}(x) = \sum_{l=0}^{n_L} \mathbb{I}_{\Omega_{Li}}(x) s_{OSRT_{Li}}(x), \quad \forall x \in \Omega \tag{4}$$

where $\mathbb{I}_{\Omega_{Li}}(x)$ is the indicator function, which is defined as

$$\mathbb{I}_{\Omega_{Li}}(x) = \begin{cases} 1 & x \in \Omega_{Li} \\ 0 & x \notin \Omega_{Li} \end{cases} \tag{5}$$

We assume that the observed datasets $f_X = \{f(x_1), f(x_2), \ldots, f(x_N)\}$ are generated by a OSRT, which we are aiming to approximate. For numerical variables, we model the generating process with online decision tree and an region-based adaptive sparse function. And we do not specify the depth of online decision tree, thus allowing us to deal with streaming data. And we will describe the details about online decision tree and the adaptive sparse function in the next section.

## 2.1 ADAPTIVE RBF EXPLORATION

In the context of approximating functions using radial basis function (RBF) networks, it is generally believed that increasing the number of neural nodes can enhance the approximation performance. However, this improvement becomes limited as the number of nodes becomes excessively large, leading to a substantial increase in computational cost and resulting in an overly complex model. Consequently, when the increase in the number of nodes no longer yields significant improvements in approximation quality, it is prudent to consider halting the expansion of the node count.The purpose of this adaptive exploration is to gradually determine the sparse and quasi-uniform centers $\succ t_q = \{\chi_1, \chi_2, ..., \chi_q\}$ of the RBF approximation $s_l$ on $\Omega_l$, this process stops when there was no significant improvement.

Suppose that we have datasets $X_l \subset \Omega_l$, and we consider a method for generating a quasi-uniform subsequence of $X_l$, which is an important step for subsequent adaptive RBF explorations. To find a quasi-uniform subsequence $\mathbb{C}$ from $X_l$, we start with the approximate mean point, that is,

$$C_1 = \arg\min_{x \in X_l} \|x - \overline{X_l}\|_2, \quad where \overline{X_l} = \frac{1}{N_l} \sum_{i=1}^{N_l} x_{li} \tag{6}$$

where $N_l$ is the data size of $X_l$. And for known $\mathbb{C}_j = \{C_1, ..., C_j\}$, the subsequent point $C_{j+1}$ is determined as

$$C_{j+1} = \arg\max_{x \in X_l}(\min_{1 <= i <= j} \|x - C_i\|_2) \tag{7}$$

$C_{j+1}$ is the point that maximizes the minimum of the set of distances from it to a point in $\mathbb{C}_j$. Notice that points in $\mathbb{C}$ are candidates for the sparse and quasi-uniform centers $\succ t_q$. By determining the distance between every point of $X_l$ and the points in $\mathbb{C}$, we can get the Voronoi diagram of $\mathbb{C}$. And we are going to add the point in $\mathbb{C}$ to $\succ t_q$ with maximum average error by the following procedure:

1) By substitute equation 1 into equation 2, we can rewrite the problem of minimizing the 2-norm of the residual as:

$$\min_{\alpha \in \mathbb{R}^{N_x}} \|r_l(X_l) - \Phi\alpha\|_2^2 \tag{8}$$

where the interpolation matrix $\Phi \in \mathbb{R}^{N_l \times N_x}$ are defined by $\Phi_{ij} = \varphi_{\delta_l}(X_{li} - \chi_j)$. Suppose $\Phi$ have a QR decomposition $\Phi = QR$, where $Q \in \mathbb{R}^{N_l \times N_x}$ has orthonormal columns and $R \in \mathbb{R}^{N_x \times N_x}$ is upper triangular, then the problem(8) has a unique solution $\alpha^* = (\Phi^T\Phi)^{-1}\Phi^T r_l(x) = R^{-1}Q^T r_l(x)$, where $R$ and $Q^T s_l(x) \in \mathbb{R}^{N_x}$ can be recursively obtained without computing $Q$ by Householder transformations(Golub & Van Loan, 2013). It means that when adding one point to $\succ t_q$, both $R_{q+1}$ and $Q_{q+1}^T r_l(X_l)$ can be recursively obtained by $R_q$ and $Q_q^T rl(X_l)$ without computing $Q_q$,

2) For a fixed factor $\theta_s$, the current shape parameter of $\varphi_{\delta_l}(x)$ can be determined as

$$\delta_l = \sqrt{-\frac{ln\theta_s}{\max\limits_{x \in X_l} \|x - \bar{X}_l\|_2^2}}, \quad \text{where} \bar{X}_l = \frac{1}{N_l}\sum_{i=1}^{N_l} X_{li}$$

Which is related to the size of domain $\Omega_l$, then a temporary RBF approximation and relevant residual can be obtained by

$$s_{t_q}(x, \alpha_{t_q}) = \sum_{i=1}^{q} \alpha_i \varphi_{\delta_l}(x - \chi_i), x \in X_l$$

where the coefficients $\alpha_{t_q} = R_q^{-1}Q_q^T r_l(x)$, and

$$r_{t_q}(x) = r_l(x) - s_{t_q}(x, \alpha_{t_q}), x \in X_l$$

3) Suppose $\{\wedge_m\}_{m=1}^{q+d+1}$ be the Voronoi diagram of $\mathbb{C}_{q+d+1}$, and $\{\wedge_m\}_{m \in \Gamma}$ are Voronoi regions with respect to those elements from the complementary set $\mathbb{C}_{q+d+1} - \succ_{t_q}$, then

$$\chi_{j+1} = C_{m^*} \in \mathbb{C}_{q+d+1} - \succ_{t_q} \tag{9}$$

where $C_{m^*}$ represents the candidate point with largest mean squared error:

$$m^* = \arg\max_{m \in \Gamma} \sum_{x \in \wedge_m \cap X_l} \frac{\|r_{t_q}(x)\|^2}{n_m}$$

and $n_m$ is the point number of $\wedge_m \cap X_l$.

4) And the termination criteria is

$$\hat{\kappa}(R) > \theta_1 \text{ or } \epsilon_q - \epsilon_{q+1} < \theta_2 \text{ or } q + 1 = N_l, \tag{10}$$

where$\hat{\kappa}(R) = \frac{\max_i |R_{ii}|}{\min_i |R_{ii}|}$ is an estimation of the condition number of R, $\theta_1$ is the upper bound of condition number of R. And $\theta_2$ is predefined parameter to ensure that the increase in the number of nodes yields significant improvements in approximation quality, and

$$\epsilon_q = \sqrt{\frac{1}{N_l}\sum_{x \in X_l}(r_{t_q}(x))^2}$$

This termination criteria is to ensure that the number of central nodes in $s_{t_q}$ is appropriate and not too many or too few.

And finally we can get the approximator $s_l$ on $\Omega_l$, and residual $r_{l+1}$ of $X_l$:

$$s_l(x, \alpha_l) = \sum_{i=1}^{N_\chi} \alpha_{li} \varphi_{\delta_l}(x - \chi_i), x \in \Omega_l \tag{11}$$

$$r_{l+1}(x) = r_l(x) - s_l(x, \alpha_l), x \in X_l \tag{12}$$

Where $N_\chi$ is the number of neurons in this node.

## 2.2 Online Node Structure optimization

Considering the model's complexity, we establish a maximum tree depth of OSRT as $d_{max}$. Once this depth is reached or the expected accuracy is reached, we cease partitioning the input space $\Omega_{d_{max}}$. To ensure continued data processing in the model, we have the capability to update the leaf nodes previously trained at the last layer of OSRT. In Sect. 2.1, we discussed how adaptive RBF exploration can be employed to determine the number and values of center points. Assume that $N_\chi$ is the number of neurons in leaf node after adaptive RBF exploration, then we set the . Consequently, when new data comes in, we can update the central node according to two principles: mean squared error and significance for the hidden neurons.

When the RBF structure is not suitable for the current data, the network residual error becomes large. At this point, any insignificant node is substituted with a fresh node. To regulate the frequency of node replacements, we employ the mean squared residual error as a metric for evaluating the RBF network's performance. In order to obtain the latest data information, we use a stack to save a fixed amount of data as $N_l$. When new data $(x, y)$ enters the stack and the amount of data exceeds the maximum value, the old data will be discarded. Define the mean squared residual error as:

$$\bar{e}^2 = \frac{1}{N_l} \cdot \frac{\sum_{x \in X_l}(r_{l+1}(x))^2}{\sum_{x \in X_l}(r_l(x))^2} \tag{13}$$

Then, we have the following criterion

$$\begin{cases} \text{if } \bar{e}^2 \leq \Delta_1, \text{ the RBF structure remains unchanged} \\ \text{if } \bar{e}^2 > \Delta_1, \text{ adding one hidden neuron to the RBF structure} \end{cases}$$

where $\Delta_1$ represents a constant threshold that's configured based on the performance requirement, and $\chi = x$ is the center point of the new added neuron. As a general rule, reducing the value of $\Delta_1$ results in a lower achievable residual error, but it may also lead to more frequent occurrences of node replacements.

We have mentioned that the maximum number of central points as $N_{max} = 1.2 N_\chi$, which means that just adding new hidden neurons is not appropriate. We need a strategy to conduct the structure pruning phase when the number of hidden neurons exceeds $N_{max}$. Here we introduce the notion of significance for the hidden neurons based on their statistical average contribution over all samples.

The network output for one sample $x_j \in X_l$ is given by

$$s_1 = s_l(x_j, \alpha_l) = \sum_{i=1}^{N_\chi} \alpha_{li} \varphi_{\delta_l}(x_j - \chi_i), x_j \in \Omega_l$$

If the neuron $k$ is removed, the output of this RBF network with the remaining $N_\chi - 1$ neurons for the input is:

$$s_2 = \sum_{i=1}^{k-1} \alpha_{li} \varphi_{\delta_l}(x_j - \chi_i) + \sum_{i=k+1}^{N_\chi} \alpha_{li} \varphi_{\delta_l}(x_j - \chi_i), x_j \in \Omega_l$$

Thus, for $x_j$, the error resulted from removing neuron $k$ is given by

$$E_{kj} = \|s_2 - s_1\| = \|\alpha_{lk}\| \varphi_{\delta_l}(x_j - \chi_k) \tag{14}$$

For datasets $X_l$, the error resulted from removing neuron $k$ is given by

$$E_k = \sqrt{\frac{1}{N_l}\sum_{j=1}^{N_l} E_{kj}^2} = \|\alpha_{lk}\|\sqrt{\frac{1}{N_l}\sum_{j=1}^{N_l}\varphi_{\delta_l}^2(x_j - \chi_k)} \qquad (15)$$

And we will finally drop the $k^*$ neuron as the pruning phase, where $k^* = \underset{k}{argmin}\, E_k,\ 1 \leq k \leq N_\chi$

## 2.3 ONLINE TREE GENERATION

For large scale problems, the conventional global RBF-based approach has prohibitive computational costs. Fortunately, the tree decomposition method offers a viable solution by yielding moderately sparse matrices, thereby mitigating the computational overhead. We consider constructing the online tree from the perspective of decision tree, in which each node contains a decision principle in form of $g(x) > \theta$. These principles usually contain two main parts (Saffari et al., 2009; Genuer et al., 2017; Zhong et al., 2020): 1) a designed function $g(x)$, which usually returns a scalar value, 2) a threshold $\theta$ which based on the datasets and function decides the left/right propagation of samples. Here we denote the function $g(x) = x \cdot v$, where $v$ is a vector related to the residual in this node.

To facilitate online learning of trees, we propose a strategy as follows: A newly constructed tree begins with a single root node, and with the streaming data arrival we accumulate online statistics. In this approach, we introduce two key hyperparameters: 1) The minimum number of samples a node must observe before considering a split, denoted as $\beta$. 2) The point at which a node's prediction accuracy plateaus, indicating that further expansion of central nodes no longer improves accuracy. Consequently, a node initiates a split when the cardinality of its dataset $N_l >= \beta_l$ and we finish the adaptive RBF exploration. In general, as the depth of a tree node increases, the represented input space becomes smaller. And the complexity of the objective function for that node decreases, which implies that the required amount of training data can be reduced as the tree depth increases. So we denoted $\beta$ as the number of samples required in the root node. For nodes in $l^{th}$ layer, the required number of samples can be expressed as:

$$\beta_l = \frac{\beta}{l}$$

Suppose we have got enough samples and done the adaptive RBF exploration on $X_l \subset \Omega_l$, where $|X_l| = N_l = \beta_l$. Then we conduct the adaptive exploration and get the approximator $s_l(x, \alpha) = \sum_{i=1}^{N_\chi}\alpha_i\theta_i(x)$. And we can express the error as this form: $r_{l+1}(x) = r_l(x) - s_l(x), \forall x \in X_l$, which will be passed to the child nodes. In order to block the spread of error, we expect to separate the points with large errors from those with small errors. First, we generate $d+1$ quasi-uniform points $\mathbb{S}_{d+1}$ of $X_l$ by the method mentioned in Sect. 2.1 with a different starting point

$$C_1 = \underset{x \in X_l}{\arg\max}\|x - \overline{X_l}\|_2, \quad where \overline{X_l} = \frac{1}{N_l}\sum_{i=1}^{N_l} x_{li}$$

Just like selecting the candidate point in $\mathbb{C}$ to $\succ t_q$ with largest mean squared error, which has mentioned in Sect. 2.1. We here also let $\{\wedge_m\}_{m=1}^{d+1}$ be the Voronoi diagram of $\mathbb{S}_{d+1}$, and determine the first splitting point of $X_l$ as:

$$x_{sp1} = S_{m^*}$$

where $m^* = \arg\max_m \sum_{x \in \wedge_m \cap X_l}\frac{\|r_{l+1}(x)\|^2}{n_m}$ and $n_m$ is the point number of $\wedge_m \cap X_l$. And the second splitting point of $X_l$ is determined as

$$x_{sp2} = \underset{x \in X_l}{\arg\max}\|x - x_{sp1}\|^2$$

Then vector $v$ mentioned above can be expressed as

$$v = x_{sp1} - x_{sp2} \qquad (16)$$

For any point $x \in X_l$, $g(x) = x \cdot v$ represents the projection length of $x$ in the direction $v$. And $g(x)$ can also be recognized as the distance from point $x$ to $x_{sp1}$, and the maximum prediction error

is around $x_{sp1}$. Let $g(X_l) = X_l \cdot v$ represents projection of $X_l$ int the direction $v$, then the threshold $\theta$ can be determined as

$$\theta = median(g(X_l)) \tag{17}$$

Then datasets $X_l$ can be splitted into $X_{l1}$ and $X_{l2}$ according to function $g(x)$ and threshold $\theta$:

$$X_{l1} = \{x \in X_l : g(x) < \theta\} \text{ and } X_{l2} = X_l - X_{l1} \tag{18}$$

And the input space $\Omega_l$ can also be splitted into $\Omega_{l_1}$ and $\Omega_{l_2}$ to guide the way for data coming in later:

$$\Omega_{l1} = \{x \in \Omega_l : g(x) < \theta\} \text{ and } \Omega_{l2} = \Omega_l - \Omega_{l1} \tag{19}$$

For each sample $(x, y)$, we will determine which child domain it belongs to according to 19. When the number of samples in the child node satisfied the required $\alpha_{l+1}$, the adaptive RBF exploration and splitting procedure will take place again. And finally OSRT will reach the leaf node which stop the splitting phase. With the arrival of new samples, we can conduct the node structure optimization mentioned in Sect. 2.2 in the leaf node.

## 2.4 TRAINING ALGORITHM OF OSRT

Here we describe the training method of OSRT as Algorithm 1. The stability properties of OSRT can be seen in Appendix A;

---

**Algorithm 1** Training of OSRT

---

**Input:** Sequential training example $(x, f(x))$; the OSRT parameters $\theta$, $\beta$, $d_{max}$ and $\Delta_1$
**Output:** OSRT
 1: Generate the root node, and set $r_0(x) = f(x)$
 2: **for** $(x_t, f(x_t)), t = 1, 2, ...$ **do**
 3:     $j = $ Findleaf$(x_t)$
 4:     Calculate $r_j(x)$ by (12);
 5:     Add $(x_t, r_j(x_t))$ to leaf node $j$;
 6:     **if** $N_j \geq \beta_j$ and $j$ is not trained **then**
 7:         Obtain $s_j(x, \alpha_j)$ on $\Omega_j$ according to adaptive RBF exploration;
 8:         Update $r_{j+1}(X_j)$ on $X_j$
 9:         **if** depth of $j \geq d_{max}$ or $r_{j+1}(X_j)$) reaches the requied accuracy **then**
10:             split $X_j$ and $\Omega_j$ into two part;
11:         **end if**
12:     **else if** $j$ is trained **then**
13:         **if** $\bar{e}^2 > \Delta_1$ **then**
14:             Add $x_t$ to $\chi$;
15:             **if** $N_\chi > N_{max}$ **then**
16:                 Calculate $E_k, 1 \leq k \leq N_{max}$ and find $k^*$;
17:                 Delete neuron $k^*$ and update $\alpha$ ;
18:             **end if**
19:         **end if**
20:     **end if**
21: **end for**
22: Finally, for all $x \in \Omega$, the OSRT prediction of the target function $f(x)$ is obtained based on (4).

---

## 3 EXPERIMENTAL RESULTS

In this section, computer simulations are given to compare the proposed OSRT algorithm with some typical online modeling approaches including GGAP RBF,,MRAN, RAN, MRLS-QPSO and APSO-SORBF algorithms. All approaches apply Gaussian nodes. In this study, we use the root mean square error(RMSE) and the mean absolute error(MAE) as a measure of accuracy:

$$RMSE = \sqrt{\frac{1}{N} \sum_{i=1}^{N} (s(x_i) - f(x_i))^2}$$

$$MAE = \frac{1}{N} \sum_{i=1}^{N} \|s(x_i) - f(x_i)\|$$

### 3.1 CHAOTIC TIME SERIES (MACKEY–GLASS) PREDICTION

The chaotic Mackey–Glass differential delay equation is recognized as one of the benchmark time series problems, which is generated from the following delay differential equation:

$$\frac{dx(t)}{dt} = \frac{ax(t-\tau)}{1+x^{10}(t-\tau)} - bx(t)$$

And we can generate the time series by the following discrete equation:

$$x(t+1) = (1-b)x(t) + \frac{ax(t-\tau)}{1+x^{10}(t-\tau)}$$

Here we let $a = 0.2$, $b = 0.1$, and $\tau = 17$, the time series is generated under the condition $x(t-\tau) = 0.3$ for $0 \le t \le \tau$ ($\tau = 17$ in our case). And we take the past four observations $[x_{t-v}, x_{t-v-6}, x_{t-v-12}, xt-v-18]$ as the input vector to predict $x_t$, here we set $v = 50$. For 4500 data samples generated from $x(t)$, the first 4000 samples were taken as training data, and the last 500 samples were used to check the proposed model.

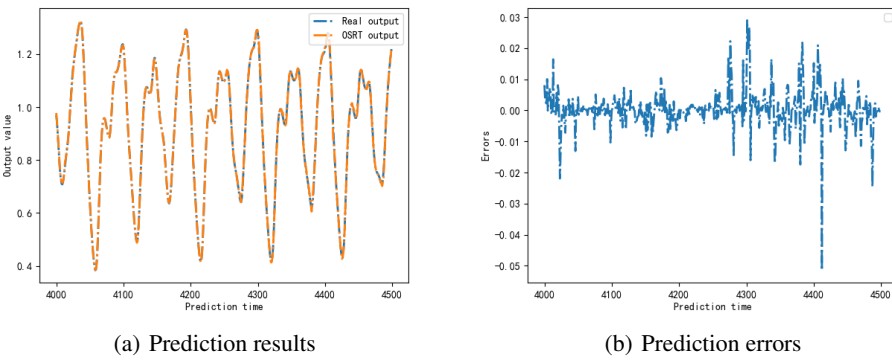

(a) Prediction results          (b) Prediction errors

Figure 1: Mackey–Glass (fixed parameters)

| Algorithms | Training RMSE | Training MAE |
|---|---|---|
| OSRT | 0.0044 | 0.0019 |
| GGAP | 0.0312 | 0.0297 |
| MRAN | 0.0337 | 0.0403 |
| MRLS-QPSO | 0.0168 | 0.0053 |
| APSO-SORBF | 0.0135 | —— |

Table 1: Mackey-Glass: Final prediction performance

Figures 1 shows the prediction results and prediction errors on the testing data, respectively. And it demonstrates that the proposed OSRT algorithm can handle this time series prediction problem very well. Table 1 shows that, compared with the other algorithms the OSRT neural network achieves the best prediction performance. Though the model size of OSRT is large than these models, but it requires the least training time because of its recursive QR decomposition tricks.

| Algorithms | Training RMSE | Training MAE |
|---|---|---|
| OSRT | 0.1071 | 0.0564 |
| GGAP | 2.3294 | 1.4829 |
| RAN | 2.7486 | 2.2543 |
| MRLS-QPSO | 0.1822 | 0.0858 |
| APSO-SORBF | 0.1726 | —— |

Table 2: Lorenz: Final prediction performance

## 3.2 LORENZ TIME SERIES PREDICTION

In this section, OSRT algorithm is applied to predict the Lorenz chaotic time series. As a 3-D and strongly nonlinear system, this time series system is governed by three differential equations as:

$$\begin{cases} \dfrac{dx}{dt} = ay - ax \\ \dfrac{dy}{dt} = cx - xz - y \\ \dfrac{dz}{dt} = xy - bz \end{cases} \tag{20}$$

where parameters $a, b, c$ control the behavior of the Lorenz system. In this simulation, the fourth-order Runge–Kutta approach with a step size of 0.01 was used to generate the system, and the parameters were set to $a = 10; b = 28; c = \frac{8}{3}$. The task in this experiment is to predict the Y-dimension samples. And we take the past four observations $[y_t, y_{t-5}, y_{t-10}, yt - 15]$ as the input vector to predict $y_{t+5}$. For 5000 data samples generated from $y(t)$, the first 2000 samples were taken as training data, and the last 3000 samples were used to check the proposed model. The results are displayed is figure 2 and table 2, and we can see that the OSRT neural network achieves the best prediction performance.

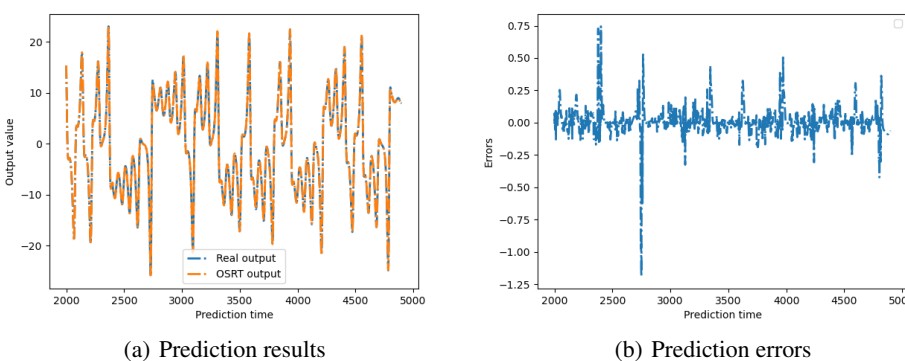

(a) Prediction results      (b) Prediction errors

Figure 2: Lorenz (fixed parameters)

## 4 CONCLUSION

An OSRT algorithm neural network for dealing with large and evolving datasets is presented in this paper. The proposed algorithm is technically built based on both online tree decomposition and online adaptive RBF explorations, which performs favorably in terms of both accuracy and efficiency. It enables the proposed OSRT algorithm to increase the tree depth and optimize the parameters of the RBF network simultaneously during the learning processes. Experimental results on the benchmark time series simulations demonstrate the efficiency and feasibility of our method. And We can also apply this method to other streaming data scenarios.

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

## A   STABILITY PROPERTIES

From Sect. 2, we have a SRT prediction of $f(x)$ on any leaf node $\Omega_L \subset \Omega$

$$s_{OSRT_L}(x) = \sum_{l=0}^{L} s_l(x), \ \forall x \in \Omega_L,$$

with the final residual

$$r_{L+1}(x) = f(x) - s_{OSRT_L}(x), \ \forall x \in \Omega_L.$$

Since $r_{l+1} = r_l - s_l$ with $r_0 = f$, then for any $1 \le l \le L$. Suppose that we are using a sparse and quasi-uniform subset $X_I \subset X_l$ to conduct the adaptive RBF exploration, then it follows from Algorithm 1 that

$$\alpha_l = R^{-1} Q^T r_l(X_I) = R^{-1} Q^T s_l(X_I),$$

where $QR$ is the QR decomposition of the current matrix $\Phi$ generated by the kernel $\varphi_{\delta_l}$ and the sparse subset $X_I$. If $\tau_l$ is the smallest singular value of $R$, then

$$\|\alpha_l\|_2 \le \tau_l^{-1} \|s_l(X_I)\|_2 \le \tau_l^{-1} \|s_l(X_l) \tag{21}$$

In addition, we have

$$(s_l(X_I), r_{l+1}(X_I))_{l_2} = 0$$

so it holds from the randomness of $X_I$ that, in expectation,

$$(s_l(X_l), r_{l+1}(X_l))_{l_2} = 0$$

According to the inclusion relationship $X_0 \supset X_l \supset X_L$ and the orthogonality of $s_l(X_l)$ and $r_{l+1}(X_l)$, we can obtain the following recurrence relations

$$\|r_l(X_l)\|_2^2 = \|r_{l+1}(X_l)\|_2^2 + \|s_l(X_l)\|_2^2, \ 0 \leq l \leq L,$$

and

$$\|r_l(X_{l-1})\|_2^2 > \|r_l(X_l)\|_2^2, \ 1 \leq l \leq L.$$

Thus, in expectation, it follows that

$$\|f_X\|_2^2 = \|s_0(X_0)\|_2^2 + \|r_1(X_0)\|_2^2 > \|s_l(X_l)\|_2^2 + \|r_{L+1}(X_L)\|_2^2.$$

Together with (21), we proved the following theorem.

**Theorem 1** *Suppose $s_{SRTL}$ is a SRT prediction of a function $f$ on a leaf node $\Omega_L \subset \Omega$ with respect to the data $(X, f_X)$, as defined in (3). Let $\alpha_l$ be the coefficients of the $l^{th}$ level least square approximation $s_l$, then in expectation,*

$$\sum_{l=0}^{L} \|\alpha_l\|_2 \leq \tau^{-1} \cdot \|f_X\|_2,$$

*where $L$ is the depth of leaf node $\Omega_L$ and $\tau = \min_{1 \leq l \leq L} \tau_l$ and the constants $\tau_l$ comes from (21).*

Note that this theorem obviously holds for our OSRT with sparsification processes introduced in Sect. 2. We shall consider functions from certain Sobolev spaces $W^{k,p}(\Omega)$ with $k \in N_0, 1 \leq p < \infty$ and native spaces of Gaussians $\mathcal{N}_\varphi(\Omega)$, respectively. The Sobolev space $W^{k,p}(\Omega)$ consists of all functions $f$ with distributional derivatives $D^\gamma f \in L_p(\Omega)$ for all $|\gamma| \leq k, \gamma \in \mathbb{N}_0^d$. And now we can prove the following theorem.

**Theorem 2** *Under the supposition of Theorem 1. For all $1 \leq p < \infty$, $k \in N_0, \delta \geq \delta_L$ and any leaf node $\Omega_L$ of the predictions OSRT, it holds that, in expectation,*

$$|s_{OSRT}|_{W^{k,p}(\Omega_L)} \leq C_W \cdot \tau^{-1} \cdot \|f_X\|_2 \text{ and } \|s_{OSRT}\|_{N_\varphi(\Omega_L)} \leq C_\mathcal{N} \cdot \tau^{-1} \cdot \|f_X\|_2,$$

*where the constant $\tau$ comes from Theorem 1, the constant $C_W$ depends on $\delta_0, \delta_L, d, p$ and $k$, and the constant $C_\mathcal{N}$ depends only on $\delta_L$ and $d$.*

**Proof** To prove the first inequality, observe that

$$|s_l|_{W_{k,p}(\Omega_L)} \leq \left( \sum_{|r|=k} \sum_j |\alpha_{l,j}|^p \|D^r \varphi_l\|_{L_p(\Omega_L)}^p \right)^{\frac{1}{p}} \leq M_{\delta_l}^{k,p} \|\alpha_l\|_p \leq C_1 M_{\delta_l}^{k,p} \|\alpha_l\|_2$$

where $M_{\delta_l}^{k,p} = \left( \sum_{|r|=k} \|D^r \varphi_{\delta_l}\|_{L_p(\mathbb{R}^d)}^p \right)^{\frac{1}{p}}$, and for any $0 \leq l \leq L$, $M_{\delta_l}^{k,p} < M_{\delta_L}^{k,p}$ when $k > 1$; or $M_{\delta_l}^{k,p} < M_{\delta_0}^{k,p}$ when $k < 1$; or $M_{\delta_l}^{k,p} = M^{k,p}$ is independent of $\delta_l$ when $k = 1$. Together with Theorem 1, we have

$$|s_{OSRT}|_{W^{k,p}(\Omega_L)} \leq \sum_{l=0}^{L} |s_l|_{W^{k,p}(\Omega_L)} \leq C_W \cdot \tau^{-1} \cdot \|f_X\|_2,$$

where $C_W = C_1 M_{\delta_L}^{k,p}$ when $k > 1$; or $C_W = C_1 M_{\delta_0}^{k,p}$ when $k < 1$; or $C_W = C_1 M^{k,p}$ when $k = 1$.

To prove the second inequality, observe that for any $s_l \in \mathcal{N}_{\varphi_{\delta_l}}(\Omega_l)$, there is a natural extension $\varepsilon_{s_l} \in \mathcal{N}_{\varphi_{\delta_l}}(\mathbb{R}^d)$ with $\|\varepsilon_{s_l}\|_{\mathcal{N}_{\varphi_{\delta_l}}(\mathbb{R}^d)} = \|s_l\|_{\mathcal{N}_{\varphi_{\delta_l}}(\Omega_l)}$. From the definition of native spaces of Gaussians, we see that $\varepsilon_{s_l} \in \mathcal{N}_{\varphi_{\delta_l}}(\mathbb{R}^d)$ with

$$\|\varepsilon_{s_l}\|_{\mathcal{N}_\varphi(\mathbb{R}^d)} \leq \|\varepsilon_{s_l}\|_{\mathcal{N}_{\varphi_{\delta_l}}(\mathbb{R}^d)} \tag{22}$$

where $\delta \geq \delta_L > \cdots > \delta_0$; and further, the restriction $\varepsilon_{s_l} | \Omega_L = s_l | \Omega_L$ of $\varepsilon_{s_l}$ to $\Omega_L \subseteq \Omega_l$ is contained in $\mathcal{N}_\varphi(\Omega_L)$ with

$$\| s_l | \Omega_L \|_{\mathcal{N}_\varphi(\Omega_L)} \leq \| \varepsilon_{s_l} \|_{\mathcal{N}_\varphi(\mathbb{R}^d)},$$

hence, we have $\| s_l | \Omega_L \|_{\mathcal{N}_\varphi(\Omega_L)} \leq \| \varepsilon_{s_l} \|_{\mathcal{N}_{\varphi_{\delta_l}}(\mathbb{R}^d)}$, and then

$$\| s_{OSRT} \|_{\mathcal{N}_\varphi(\Omega_L)} \leq \sum_{l=0}^{L} \| s_l | \Omega_D \|_{\mathcal{N}_\varphi(\Omega_L)} \leq \sum_{l=0}^{L} \| \varepsilon_{s_l} \|_{\mathcal{N}_{\varphi_{\delta_l}}(\mathbb{R}_d)}$$

Together with Theorem 1 and

$$\| \varepsilon_{s_l} \|_{\mathcal{N}_{\varphi_{\delta_l}}(\mathbb{R}_d)}^2 = \int_{\mathbb{R}^d} |\hat{s}_l(\omega)|^2 e^{\frac{\|\omega\|_2^2}{4\delta_l^2}} d\omega \leq \| \alpha_l \|_1^2 \int_{\mathbb{R}^d} e^{-\frac{\|\omega\|_2^2}{4\delta_l^2}} d\omega \leq C_2^2 (2\delta_L)^d \pi^{d/2} \| \alpha_l \|_2^2$$

we finally have $\| s_{OSRT} \|_{N_\varphi(\Omega_L)} \leq C_{\mathcal{N}} \cdot \tau^{-1} \cdot \| f_X \|_2$, where $C_{\mathcal{N}} = C_2 (2\delta_L)^d \pi^{d/2}$.

