# OpenReview forum: "OSRT: An Online Sparse Approximation Model for Scattered Data"
_ICLR.cc/2024/Conference — Submitted to ICLR 2024_

### Official Review · Reviewer_TLX1 · 2023-10-30

**Soundness:** 2 fair
**Presentation:** 2 fair
**Contribution:** 2 fair
**Rating:** 5
**Confidence:** 3

**Summary:**

This paper proposes a method, Online Sparse Residual Tree (OSRT) for
handling streaming multivariate scattered data. The proposed
method is built on the sparse residual tree (SRT) method proposed in [Xu & Luo, 2022] and extended to deal with
evolving data efficiently in an online fashion.

The proposed OSRT model dynamically updates the tree structure by adding or deleting neurons and by splitting nodes as a new training sample arrives.
Experiments demonstrate that the ORST method has superior performance to other online algorithms.

**Strengths:**

- With the proposed online extension, the SRT framework can now learn streaming data in an online fashion to predict future data.
- The experiments demonstrate the proposed method outperforms the state-of-the-art base-line methods in the literature.

**Weaknesses:**

- There are some imprecise parts which make it difficult to evaluate the feasibility of the proposed method. For example, in Section 2.2, on page 5, the sentence "then we set the." is incomplete. Algorithm 1 is not fully explained in the text. For example, FindLeaf() in step 3 is not defined in the text. The step 9 seems to contradict what they say in the text. I supporse if the condition is NOT satisfied then it should do splitting. On page 5, the authors state that "We have mentioned ... as $N_{max} = 1.2 N_{\chi}$," but they never mentioned it earlier.
- The SRT, which is the previous work, is treated as if originally proposed in this paper. The authors should clearly split Section 2 into two separate sections, one for explaining the previous SRT as background and the other for the proposed online extensions.
- The details of the hyperparameter settings used in the experiments are missing completely. The hyperparameters include the maximum tree depth $d_{max}$, the factor $\theta_s$, the stack size $N_l$ and the error threshod $\Delta_1$. Changing their values may influence their performance and setting them to appropriate values may be non-trivial. However, none of their concrete values nor
their robustness to the performance in the experiments is reported.

**Questions:**

Because OSRT is an extension of SRT, I would like to know the performance difference
between the original SRT and its online version OSRT. The ORST is an online algorithm and evaluates each
sample only once according to Algorithm 1 on page 7. Therefore
some performance degradation is expected against SRT, while OSRT is more
computationally efficient. The extent of the performance degradation is important
information to understand the potential of the proposed method and should be reported.

Minor comments:

In Section 2 on page 2, the Gaussian kernel is defined as $\theta_j(x)$ that includes $c_j$ as its center vector but a different
symbol $\phi_j(x)$ is used in the following equation.
On page 4, $\phi_{\delta_l}(X_{li} -\chi_j)$ is used, where the definition of $\phi_{\delta_l}(x)$ does not include $c_j$ and the suffix of $\phi_{\delta_l}$ is the shape parameter, while the suffix of $\phi_j$ is the node index.

On page 4, $\sum_{i=1}^{t_q}$ should be $\sum_{i=1}^{q}$.

In Section 2.1, $\prec t_q$ is defined but $t_q$ is not defined at all and is still used in a couple of places.

In Equation (8), the notation $r_l(x)$ is misleading. It should be $r_l(X_l)$ as  used
later in $Q^T_{q+1}r_l(X_l)$.

In Section 2.3 on page 6, the definition of $S_m$ is unclear. $S_m$ is supposed to be a vertex of Voronoi diagram.

The right hand side of Equiation (1) : $\sum_{i=1}^{N_{\chi}} \alpha_i \phi_{\delta_l}(x)$ is confusing because
$\sum_{i=1}^{N_{\chi}} \alpha_i \phi_{\delta_l}(x) = \phi_{\delta_l}(x)\sum_{i=1}^{N_{\chi}} \alpha_i $

**Details Of Ethics Concerns:**

I have no concerns.

---

### Official Review · Reviewer_4t5w · 2023-10-31

**Soundness:** 2 fair
**Presentation:** 2 fair
**Contribution:** 1 poor
**Rating:** 3
**Confidence:** 4

**Summary:**

The paper extends methods for radial basis function (RBF) neural networks to predict time-series to online models---named an "online sparse residual tree" (OSRT) model. OSRT involves building a sparse tree in which the RBF networks reside.  To address streaming time-series, the model's online adaptation is done by thresholding the current mean squared residual error as new data arrives.

**Strengths:**

The paper presents several novel ideas, by combining RBF networks, sparse regressison trees, and online updating for time series prediction.

**Weaknesses:**

The paper lacks a principled approach to model design and evaluation, appearing to have little rationale in the combination of methods used beyond their adaptation from the recent literature, and their apparent heuristic value.  Typically one would expect a cross validation step as part of the algorithm when complexity parameters or thresholds are called for in a model.

The exposition is hard to follow at best, and at times incomplete, or the symbols are incorrect.
For instance, in Section 2:

- Gaussian kernel is designated \theta, but in the approximation the character \phi is used -- are these the same thing? Note that \theta is reused with a different meaning in Equation (17).

- After equation (2) the phase "Where Nχ is the number of neurons in this node, δl is the shape parameter." makes no sense since neither variables appear in the previous formula.  The rest of that paragraph has similar problems with reference to variables not introduced in the equations that it attempts to explain.

In general, one needs a principled method for determining the complexity of the model, e.g. the number of nodes, such as cross validation, or use of complexity penalty terms, e.g. in BIC. Is the maximum tree depth (Section 2.2) something one calculates, or is it a parameter one setd? Simply considering when "the increase in the number of nodes no longer yields significant improvements in approximation quality" will lead to overfitting. "Significant improvement" is not a principled method.

**Questions:**

There are many terms introduced in the explanation of the model that are introduced but not explained:  Could you describe the tree in terms of its layers?  What is the "split rule" and what is the stopping condition referred in the paragraph following Equation (2)?  In what sense is it sparse? Is there a sparsification step?  What do you mean in Section 2.1 by "quasi-uniform"?  Are your "mean points" C the same as your centers? Honestly this as far as I got in the text.

---

### Official Review · Reviewer_nWJS · 2023-11-03

**Soundness:** 2 fair
**Presentation:** 1 poor
**Contribution:** 2 fair
**Rating:** 5
**Confidence:** 3

**Summary:**

This paper presents a predictive statistical model OSRT for handling streaming multivariate scattered data. The OSRT model can dynamically expand its network depth with the arrival of data. A RBS refinement is also incorporated into the OSRT model to minimize its residual error. Moreover, the paper proposes an incremental method to explore the central node of the RBF function, ensuring the sparsity and accuracy of the model. Theoretical analysis and Empirical results are provided to demonstrate the effectiveness of the proposed OSRT mode.

**Strengths:**

S1. The paper focuses on online regression analysis, which is an important problem especially considering the growing necessity to process large-scale data in the era of Big Data.

S2. The paper proposes several approaches to minimize the residual error. The effectiveness of the proposed method is theoretically proved and empirically demonstrated.

**Weaknesses:**

My main concern is the presentation of the paper.

1. There is no formal problem definition in the introduction, which makes it almost impossible for non-experts to understand the paper.

2. The introduction part is too short and not very informative. The authors should at least illustrate some of the backgrounds of online regression analysis and highlight existing challenges.

3. The authors did not clearly state the technical contributions of the work. The related work part is also messy, which makes it very hard for me to identify the contributions of the paper.

4. the author did not present any intuition for the proofs, which makes it hard to verify the correctness.

5. the current manuscript contains numerous typos, unclear sentences, and undefined notations. For instance:

- Page 1: For example, The partition

- Page 1: with more and more data is generated

- Page 1: have deriving

- Page 1: too large a network may bring in ...

- Page 1: takes the growing strategy first, it adds

- Page 2: It separate

- Page 2: represented blow

- Page 2: Where

- Page 3: Where

- Page 3: Most regression trees grown by

- Page 3: $r_{l+1, j}$ combined into

- Page 3: the notation $\varphi$ requires clarifications

- Page 3: $i \neq j$ Then -> $i \neq j$. Then

- Equation (4): $\mathbb{I}$ and $1_{\Omega_{L_i}}$

- Page 4: then the problem (??)


In general, I think the paper is promising. However, the presentation of the paper does not meet the high standards of ICLR.

**Questions:**

Please refer to the Weaknesses part for details.

---

### Meta-Review · Area_Chair_C2Xb · 2023-12-07

**Metareview:**

This paper presents a novel predictive statistical model, the Online Sparse Residual Tree (OSRT), for handling online multivariate scattered data. The proposed method is built on the sparse residual tree (SRT) method, and extended to deal with evolving data efficiently in an online fashion. The proposed method dynamically updates the tree structure by adding or deleting neurons and by splitting nodes as a new training sample arrives. Theoretical analysis and Empirical results are provided to show the effectiveness of the proposed method.

Firstly, the studied problem is an important one for big data analysis. Secondly, the combination method is somehow novel. Finally, the experiments show satisfactory performance on some datasets.

However, the novelty of this paper is limited, since the combination is not in a principled way. Secondly, the paper should use a more principled method to determine the hyperparameters. Finally, the paper should be polished more carefully.

**Justification For Why Not Higher Score:**

The paper should use a more principled way to explain the combination, to determine the hyperparameters.

**Justification For Why Not Lower Score:**

N/A

---

### Decision · Program_Chairs · 2024-01-16

Reject